

# Evaluating generative AI integration in Saudi Arabian education: a mixed-methods study

Abdullah Alammari

Faculty of Education, Curriculums and Teaching Department, Umm Al-Qura University, Makkah, Makkah, Saudi Arabia

## ABSTRACT

Incorporating generative artificial intelligence (GAI) in education has become crucial in contemporary educational environments. This research article thoroughly investigates the ramifications of implementing GAI in the higher education context of Saudi Arabia, employing a blend of quantitative and qualitative research approaches. Survey-based quantitative data reveals a noteworthy correlation between educators' awareness of GAI and the frequency of its application. Notably, around half of the surveyed educators are at stages characterized by understanding and familiarity with GAI integration, indicating a tangible readiness for its adoption. Moreover, the study's quantitative findings underscore the perceived value and ease associated with integrating GAI, thus reinforcing the assumption that educators are motivated and inclined to integrate GAI tools like ChatGPT into their teaching methodologies. In addition to the quantitative analysis, qualitative insights from in-depth interviews with educators unveil a rich tapestry of perspectives. The qualitative data emphasizes GAI's role as a catalyst for collaborative learning, contributing to professional development, and fostering innovative teaching practices.

# INTRODUCTION

Since the 1980s, the emergence of artificial intelligence in education has signaled a transformative period in the field. It reflects the growing fascination among scholars with the fusion of technology and pedagogy (*Sleeman & Brown, 1982*). According to contemporary perspectives, artificial intelligence (AI) extends beyond mere technology, encompassing machine learning, neural networks, natural language processing, and various methods for replicating human cognitive functions (*Baker, Smith & Anissa, 2019*; *Akgun & Greenhow, 2022*). *Luckin (2017)* delved into the potential of artificial intelligence in education (AIEd), emphasizing its capacity to enhance learning outcomes through personalized, flexible, and engaging educational experiences. AIEd not only benefits students but also aids educators in offering tailored support (*Dolmark et al., 2021*, *2022*).

Machine learning, often abbreviated as ML, forms the foundation for both categories of artificial intelligence (AI). When integrated with other technologies, it can augment its capabilities and reduce its dependence on human guidance, as noted by *Akgun & Greenhow (2022)*. ML algorithms have found diverse applications, such as delivering product or service recommendations, recognizing faces, predicting academic performance,

Corresponding author
Abdullah Alammari,
amammari@uqu.edu.sa

and detecting diseases (*Kabudi, Pappas & Olsen, 2021*; *Zhai et al., 2021*). Furthermore, *Luckin (2017)* stresses the significance of AI in real-world learning materials, as it provides assessments, insights into student progress, and addresses achievement gaps. Ultimately, *Luckin (2017)* assert that the benefits of AI in education outweigh the challenges, making it a worthwhile pursuit.

Various capabilities have gradually evolved in artificial intelligence-driven by machine learning. Among these capabilities is generative AI (GAI), whose roots trace back to the 1960s. However, it was not until approximately 2014 that researchers notably focused on exploring GAI, as *Gui et al.*'s *(2023)* research findings underscored. GAI is broadly defined as AI that can generate new content based on input (*Pavlik, 2023*). Previous studies emphasize the significance of gaining insights into teachers' viewpoints regarding implementing emerging technologies (*Qahl & Sohaib, 2023*), as exemplified by extended reality (*Kaplan-Rakowski et al., 2023*) and virtual reality (*Kaplan-Rakowski, Dhimolea & Khukalenko, 2023*). However, general artificial intelligence (GAI) integration has received limited attention in the literature, with a few exceptions (*Kuleto et al., 2022*). Despite the increasing popularity and accessibility of GAI tools, which could be attributed to their relatively recent emergence, there remains a dearth of knowledge concerning teachers' perceptions of integrating GAI, such as Chat GPT, into educational settings (*Celik et al., 2022*; *Wang, Liu & Tu, 2021*). Previous research suggests that the adoption and proliferation of technology heavily hinge upon teachers' perspectives (*Kaplan-Rakowski et al., 2023*; *Ismail, Almekhlafi & Al-Mekhlafy, 2010*). Omitting the initial exploration of teachers' attitudes towards GAI technology could jeopardize the successful incorporation of such technology in education. Therefore, this study proposes that AI in education (AIEd) can bolster teacher preparedness. Consequently, this research builds upon the work of *Kaplan-Rakowski et al. (2023)* and is driven by the following research questions:

- What are the Saudi educators' perceptions regarding integrating GAI in the Saudi education system?
- Does the level of educator integration with GAI technology significantly predict the frequency of GAI utilization in educational settings?

The article follows a structured format, with "Literature Review" providing a comprehensive literature review to establish the theoretical background and context. "Methodology" outlines the methodology employed in the research. In "Results", both quantitative and qualitative results are presented and analyzed, shedding light on the empirical findings. "Discussions" delves into the interpretation and discussion of the results, offering insights, implications, and potential future research directions. Finally, the article concludes in the last section, summarizing the essential findings and their significance within the broader field of study. This structured approach ensures a clear and systematic presentation of the research process and outcomes.

## LITERATURE REVIEW

### AI in education

As noted by *Popenici & Kerr (2017)*, AI holds the potential to usher in transformative changes within the realm of education. Its capacity to revolutionize both the learning process for students and the teaching methods employed by educational institutions is particularly noteworthy. For example, Georgia Tech in the United States introduced a virtual teaching assistant powered by IBM's Watson platform, earning acclaim from students (*Maderer, 2016*). This instance underscores the potential of AI to enhance the teaching process, especially when dealing with student populations and facilitating personalized student interactions. Furthermore, the artificial intelligence in education (AIEd) field, as highlighted by *Schiff (2021)*, is rapidly evolving to leverage innovative software and hardware tools to enhance the quality and effectiveness of teaching and learning. AIEd encompasses a range of application areas, including intelligent tutoring systems, educational agents, adaptive assessments, educational robots, and lifelong intelligent mentors (*Schiff, 2021*). It is widely believed that modern AIEd has the potential to bridge the quality gaps exposed by previous educational technologies, as it aspires to replicate the role of educators. Through AIEd tools, teachers can shift their focus to higher-level tasks such as curriculum design and assessment while students receive tailored instruction tailored to their needs and learning styles (*Schiff, 2021*).

Nonetheless, integrating artificial intelligence into the classroom poses several challenges. Researchers have raised ethical concerns regarding data use, algorithm transparency, and potential biases, including worries about data privacy, transparency in algorithmic decision-making, and inherent biases within algorithms (*Southgate, Smith & Scevak, 2019*; *Sohaib & Olszak, 2021*; *Akgun & Greenhow, 2022*). Another significant issue is algorithmic transparency, a known challenge in the AI community, as many AI systems operate as "black boxes," rendering their decision-making processes inscrutable to humans (*Burrell, 2016*). The absence of transparency can lead to trust issues for educators and students, mainly if their assessments or recommendations are influenced by AI system decisions (*Burrell, 2016*). Extensive research on integrating AI into education, as emphasized by *Popenici & Kerr (2017)*, suggests that it can significantly influence the governance and structure of academic institutions.

Furthermore, there are potential risks associated with using AIEd in the classroom, including privacy concerns, biases, and the possibility of technology entirely replacing human educators (*Schiff, 2021*). Therefore, responsible research is crucial to address these risks, as *Schiff (2021)* underscores. Integrating social responsibility into processes and cultures is also imperative in addressing these concerns. Additionally, AI can perpetuate biases, leading to unfair or discriminatory outcomes stemming from biases in training data and inherent AI biases (*Benjamin, 2020*). For instance, some AI systems may inadvertently favor students from specific socioeconomic backgrounds, raising equity and fairness concerns in education (*Benjamin, 2020*).

Consequently, adopting AI in educational settings must be cautiously and carefully considered to mitigate these concerns (*Zawacki-Richter et al., 2019*). AI can provide

tailored solutions to cater to the support requirements of a diverse range of students (*Popenici & Kerr, 2017*). Additionally, AIEd aids in fostering learning and helping students overcome challenges related to communication and teamwork, skills vital for their holistic growth (*McLaren, Scheuer & Mikšátko, 2010*).

Artificial intelligence in education (AIEd) is increasingly recognized as a solution to teaching and learning challenges. One of the educational approaches it offers is individualized tutoring, providing flexibility, personalization, inclusivity, and effectiveness similar to one-on-one instruction (*Zanetti, Iseppi & Cassese, 2019*). These innovative tools aim to engage students and improve lessons by integrating practical, attentive, and perceptual user interfaces. AIEd tools can even analyze students' facial expressions and reactions (*Chaudhri et al., 2013*), a valuable capability, particularly in primary education, where understanding and responding to students' emotional responses can be beneficial, especially during challenging times (*Zanetti, Iseppi & Cassese, 2019*). Moreover, AIEd offers intelligent tutoring systems (ITS) that simulate a personalized one-on-one tutoring experience, considering learners' specific needs (*McLaren, Scheuer & Mikšátko, 2010*). These systems provide activities and feedback, allowing learners some control over their learning process and fostering the development of self-regulation skills (*McLaren, Scheuer & Mikšátko, 2010*). This is particularly advantageous in education, where nurturing self-regulation skills is an essential learning objective.

In summary, the literature review provides an in-depth exploration of AIEd, offering a comprehensive understanding of its potential benefits and associated challenges. *Popenici & Kerr (2017)* and *Schiff (2021)* highlight AI's transformative capacity in revolutionizing learning and teaching methods, showcasing innovative tools for personalized instruction. Ethical concerns, algorithm transparency challenges, and potential biases are raised by multiple researchers (*Southgate, Smith & Scevak, 2019*; *Sohaib & Olszak, 2021*; *Akgun & Greenhow, 2022*). The need for responsible research and social responsibility integration in AIEd is emphasized by *Schiff (2021)*. *McLaren, Scheuer & Mikšátko (2010)* and *Zanetti, Iseppi & Cassese (2019)* underscore the role of AIEd in fostering learning, overcoming challenges, and providing individualized tutoring experiences. *Chaudhri et al. (2013)* discusses the engaging and perceptual user interfaces of AIEd tools. The literature collectively points to the potential risks and rewards associated with AIEd, emphasizing the need for careful consideration and responsible adoption in educational settings.

## Generative AI—what is ChatGPT?

The availability of AI tools like ChatGPT in 2022 has brought AI into the spotlight, making society more aware of its existence and its possible consequences on how we go about our daily activities (*Lampropoulos, Ferdig & Kaplan-Rakowski, 2023*). Stable Diffusion and DALL-E have enabled the generation of images and videos from text inputs. ChatGPT, a generative pre-trained Transformer, is capable of text generation, language translation tasks and summarization. Furthermore, ChatGPT can provide detailed responses to user queries like text and code. The advanced results generated by AI have prompted users to recognize generative AI tools as valuable assistants in problem-solving and content creation. However, they have also voiced apprehensions regarding potentially diminishing

human creativity and academic integrity (*Ali, 2021*; *Schiff, 2021*; *Cope & Kalantzis, 2021*; *Sharples, 2022*).

# METHODOLOGY

This study employs a mixed-method approach, combining quantitative data collection through surveys and qualitative data collection through interviews to gather insights from educators in Saudi institutions. The Triangulation Design, depicted in Fig. 1, is the most prevalent and widely recognized approach to integrating research methods (*Creswell et al., 2003*).

## Quantitative method

This study utilized an online survey instrument partially derived from a previously validated survey developed by *Kaplan-Rakowski et al. (2023)* and *Wozney, Venkatesh & Abrami (2006)*. Participants were required first to review and acknowledge the consent form, confirming their eligibility as educators who had utilized ChatGPT at least once. Initially, participants were tasked with selecting the technology integration stage (out of six options: awareness, learning, understanding, familiarity, adaptation, and creative application) that best described their progression with general artificial intelligence (GAI). Following this, participants were required to rate their level of agreement, using a scale from 1 (strongly disagree) to 6 (strongly agree), in response to statements related to their perceptions of GAI implementation in education. To suit the specific context of this study on GAI technology, 15 items and certain statements were adapted from *Kaplan-Rakowski et al. (2023)* and *Wozney, Venkatesh & Abrami (2006)*.

Also, experts in Saudi higher education specializing in educational technology assessed and provided feedback on the survey items' operationalization. After making necessary revisions based on their input, the expert panel confirmed the instrument's content validity. In evaluating generative AI integration in Saudi Arabian education, participants were systematically chosen and recruited to ensure a representative sample reflective of the targeted educational context. The recruitment process involved reaching out to educators within Saudi higher institutions through collaboration with educational authorities and institutions. The specific criteria for determining participation eligibility included individuals actively engaged in teaching roles within Saudi higher education settings. This criterion aimed to capture insights from educators directly involved in the learning and instructional processes impacted by generative AI integration.

Additionally, participants were required to have a minimum level of familiarity with AI technologies to provide meaningful perspectives on the subject matter. A total of 140 participants responded to the survey. The participants were educators in Saudi higher institutions. After removing incomplete or missing data, 125 were used for the analysis.

## Qualitative method

A thorough and nuanced research approach is necessary to understand how artificial intelligence (AI) in education (AIEd) is implemented in the Saudi education system. This study employs the qualitative phenomenological method in our study, a method

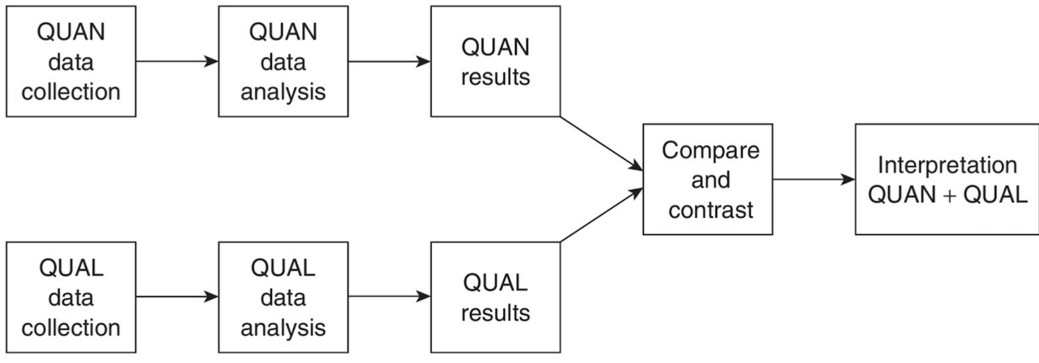

**Figure 1 Research design.** Triangulation research design.

recommended by Creswell in 2003, which delves into individuals' experiences and perceptions (*Creswell et al., 2003*). This study uses a qualitative phenomenological approach to uncover the intricate layers of experiences and perceptions held by primary school teachers in Saudi concerning the incorporation of AIEd into their teaching practices. More specifically, this study employed the thematic analysis method alongside the phenomenological approach. This combination helps us carefully scrutinize the data we have collected during our research. The thematic analysis enables us to explore, identify, articulate, and structure the underlying themes and patterns within our data, as suggested by *Nowell et al. (2017)*.

In this study, qualitative data was collected by conducting detailed interviews with three experienced educators in higher education in Saudi Arabia, referred to as P1, P2, and P3. This study gathered qualitative data through in-depth interviews with three highly experienced educators in Saudi Arabia, identified as P1, P2, and P3, all holding Ph.D. degrees and offering diverse perspectives from their respective specializations in higher education. P1 brings expertise in education technology, P2 contributes insights from computer science, and P3, a recognized leader in the academic community, offers perspectives on engineering. These educators were selected to provide a comprehensive understanding of the implications of Generative AI integration in Saudi Arabian higher education, bringing a wealth of knowledge and diverse experiences to enrich the study's qualitative findings. Our goal with these interviews was to gain insight into educators' perspectives, observations, and insights regarding integrating artificial intelligence in education. The interviews were carried out in the English language and were meticulously transcribed *verbatim* to facilitate further analysis.

## RESULTS
### Educators' perceptions of GAI integration in education
*Quantitative findings*
To answer the first research question: What are the Saudi educators' perceptions regarding integrating GAI in the Saudi education system? Fifteen questions assessed educators' perspectives regarding applying GAI in education. Survey participants were tasked with expressing their degree of agreement or disagreement with these statements on a six-point

scale, where a score of 1 denoted "strongly disagree," and a score of 6 indicated "strongly agree." To ensure the reliability of this new measurement tool, its internal consistency was assessed using the Spearman-Brown stepped-up coefficient, which was found to be satisfactory ($\alpha$ = 0.78). Table 1 shows the results.

The assessment outcomes demonstrate a range of perspectives from the participants regarding technology's role in education. Participants generally expressed favorable views regarding its impact on academic progress (with an average score of 3.79 and a standard deviation 1.30). However, there was some diversity in their feedback. Conversely, concerns about technology potentially diverting students from traditional learning methods received a less favorable average rating (2.82, with a standard deviation of 1.21), indicating reservations among the respondents. However, participants generally exhibited confidence in the effectiveness of technology, particularly when they believed they could implement it successfully (with an average score of 4.15 and a standard deviation of 1.18). The influence of technology on student collaboration (average score of 3.00 and standard deviation of 1.24) and the development of communication skills (average score of 3.14 and standard deviation of 1.41) generated mixed feedback, reflecting varying perspectives on their impact. Overall, participants perceived technology as a valuable instructional tool (average score of 4.10 and standard deviation of 1.10) and believed it contributed to their professional development (average score of 4.25 and standard deviation of 1.23). However, concerns arose regarding potential increases in student stress and anxiety (average score of 3.80 and standard deviation of 1.29) and additional planning time (average score of 3.58 and standard deviation of 1.36). While technology was seen as a means to motivate student engagement in learning activities (average score of 3.71 and standard deviation of 1.26), the notion of it potentially impacting the number of educators in the future (average score of 4.15 and standard deviation of 1.42) was approached with careful consideration. The data reveals a multifaceted portrayal of technology's part in education, acknowledging both prospects and challenges from the participants' perspectives.

### Qualitative findings

Findings from the Interviews show similar themes, such as P1: "*I believe there are both positive and negative aspects of using GAI in education. Sometimes, I find it extremely beneficial; it's truly an incredible tool. However, when it comes to university-level work, I have concerns. In the recent term, I observed many university students utilizing AI technology like Excel for their writing, leading to potential plagiarism issues*".

P2 teacher expressed, "*I've seen a noticeable improvement in students' academic performance since we introduced GAI in our curriculum. It provides personalized assistance and feedback, helping students grasp concepts better and ultimately leading to higher grades.*" P3 shared concerns: "*I've noticed that some students tend to rely solely on GAI for information, neglecting valuable resources like library books. It's essential to strike a balance and encourage them to use traditional and AI-based sources.*" A different perspective came from a P1 who said, "*GAI has been a catalyst for collaborative learning in my classroom. Students are working together on projects, discussing their findings from AI tools, and it's fostering a sense of teamwork and knowledge sharing that I hadn't seen to this extent before.*"

**Table 1 Educators' perceptions of GAI integration in education (average score and standard deviation), first column of the table presents statement, second column presents average score, third column presents standard deviation.**

| Statement | M | SD |
|---|---|---|
| 1. Increases academic achievement (*e.g.*, grades). | 3.79 | 1.30 |
| 2. Results in students neglecting important traditional learning resources (*e.g.*, library books). | 2.82 | 1.21 |
| 3. Is effective because I believe I can implement it successfully. | 4.15 | 1.18 |
| 4. Promotes student collaboration. | 3.00 | 1.24 |
| 5. Promotes the development of communication skills (*e.g.*, writing skills, presentation skills). | 3.14 | 1.41 |
| 6. Is a valuable instructional tool. | 4.10 | 1.10 |
| 7. Makes teachers feel more competent as educators. | 3.62 | 1.25 |
| 8. Is an effective tool for students of all abilities. | 3.74 | 1.34 |
| 9. Enhances my professional development. | 4.25 | 1.23 |
| 10. Eases the pressure on me as a teacher. | 3.67 | 1.40 |
| 11. Motivates students to get more involved in learning activities. | 3.71 | 1.26 |
| 12. Should reduce the number of teachers employed in the future. | 4.15 | 1.42 |
| 13. Will increase the amount of stress and anxiety students experience. | 3.80 | 1.29 |
| 14. Requires extra time to plan learning activities. | 3.58 | 1.36 |
| 15. Improves student learning of critical concepts and ideas. | 3.68 | 1.32 |
| Average: | 3.60 | 1.30 |

P2 reflected on the impact of GAI on their growth, saying, "*Personally, I've found that integrating GAI into my teaching has been a significant boon for my professional development. It has forced me to adapt, learn new technology, and explore innovative teaching methods, which has been a fulfilling and intellectually stimulating journey.*" P3 shared his perspective on the potential implications, stating, "*I'm a bit concerned that the increasing use of GAI might reduce the number of teaching positions. While it can be efficient in some tasks, it can't replace human educators' guidance and mentorship. We need to be cautious about striking the right balance.*"

Participants shed light on their perspectives and experiences, providing valuable qualitative insights into their views on various aspects of the subject matter. Participant P1 articulated a generally positive stance regarding the integration of GAI in education. They emphasized the effectiveness of technology when it aligns with their belief in successful implementation. This participant exhibited enthusiasm for technology's potential to enhance academic achievement. However, they also expressed concerns about the impact on traditional learning resources, indicating reservations about students potentially neglecting these valuable sources of knowledge. Moreover, P1 acknowledged the role of technology in promoting student collaboration but noted mixed feedback on its effectiveness in developing communication skills. Their responses suggested that technology is viewed as a valuable instructional tool, contributing to professional development and student engagement motivation.

In contrast, participant P2 presented a more cautious perspective regarding technology's role in education. While acknowledging its potential to enhance academic

achievement, P2 expressed reservations about the potential consequences of technology's integration, particularly concerning students needing to pay more attention to traditional learning resources. This participant was optimistic about the value of GAI as an instructional tool but voiced concerns about increased stress and anxiety among students and the additional time required for planning. P2 held a balanced view regarding technology's potential to reduce the number of teachers in the future, acknowledging the topic with caution.

The examination of interviews with educators in Saudi Arabia unveils several central themes associated with incorporating GAI in educational settings. Respondents generally recognized the positive aspects of GAI, emphasizing advantages such as enhanced academic performance, personalized support, and collaborative learning opportunities. However, concerns were voiced regarding potential adverse outcomes, including issues related to plagiarism, overreliance on GAI for information by students, and the potential reduction of teaching positions. A recurring theme underscored the necessity for a balanced approach, promoting utilizing both traditional and AI-based sources. Additionally, participants deliberated on GAI's impact on professional development, emphasizing the need to adapt and acquire proficiency in new technologies within the educational landscape. In summary, the identified themes encompass the dual nature of GAI's impact, addressing both its benefits and challenges within the realm of higher education in Saudi Arabia.

In summary, the qualitative analysis of interviews with participants highlights a diversity of views on integrating GAI in education. While P1 leans towards optimism and emphasizes the positive aspects, P2 adopts a more cautious stance, emphasizing potential drawbacks and challenges. These interviews reveal the complexity of the subject and the varied experiences and perspectives of the participants. These responses show a range of opinions and experiences, indicating the multifaceted impact of GAI in education, from positive academic outcomes to concerns about overreliance and the promotion of collaboration among students.

## The relation between GAI integration and educators' frequency of GAI use

To answer the second research question: Does the level of educator integration with GAI technology significantly predict the frequency of GAI utilization in educational settings? The participant's educators' level of GAI integration was analyzed from a range of six categories, as adapted from the work of *Wozney, Venkatesh & Abrami (2006)* and *Kaplan-Rakowski et al. (2023)*.

- Awareness: Acknowledging the existence of GAI technology but still needing to utilize it, perhaps due to apprehension.
- Learning: Actively acquiring foundational knowledge about GAI, occasionally experiencing frustration and lacking confidence.
- Understanding: Understanding how to use GAI like ChatGPT and identify specific applications.

- Familiarity: Gaining self-assurance in using GAI for particular tasks and feeling comfortable with it.
- Adaptation: Consider GAI a valuable instructional tool and no longer consider it a technological challenge.
- Creative application: Proficiently integrate ChatGPT into the curriculum and use it as an instructional aid.

Analysis of the quantitative data revealed that the majority of participants (approximately 62%) fell into the third stage (Understanding) and fourth stage (Familiarity) of GAI integration. Approximately 10% of participants reported being at the fifth stage (Adaptation), and about 15% were at the most advanced stage (Creative Application). The two initial phases of integration (Awareness and Learning) were represented by approximately 14% of participants. The frequency of GAI utilization in teaching among the respondents is categorized as follows: never (approximately 40%), rarely (about 25%), when necessary (roughly 15%), often (approximately 15%), and always (around 5%).

A linear regression is used to predict GAI utilization based on educator level of integration.

*GAI Utilization = a + b * Educator Level of Integration*

Table 2 shows how the "educator level of integration" influences GAI utilization.

The regression analysis results indicate a statistically significant relationship between the two variables under investigation. The moderate multiple R-value of 0.52 suggests this relationship. The R square value of 0.279 implies that roughly 27.9% of the variation in GAI Utilization can be attributed to the independent variable, which appears to influence GAI Utilization. The ANOVA table reaffirms the significance of the regression, with a high F-statistic of 26.03 and an extremely low $p$-value (2.9719E−06). The coefficient for GAI Utilization is 0.628, with a very low $p$-value (2.97E−06), indicating its strong positive impact on the dependent variable. In summary, these findings suggest a substantial and positive relationship between the independent variable and GAI utilization, with the model being a good fit for the data.

## DISCUSSIONS

### Quantitative insights

With the integration of AI, the educational landscape is on the brink of a significant transformation characterized by enhancing personalized learning experiences and automating administrative tasks (*Southgate, Smith & Scevak, 2019*; *Zulkarnain & Yunus, 2023*). It is essential to recognize that AI holds substantial potential in bridging achievement gaps and furnishing students and educators with customized support that aligns with their specific requirements (*Luckin, 2017*). However, it is imperative to acknowledge the existence of challenges and apprehensions alongside these promising prospects, which necessitate attention. Among the vital areas that educators and policymakers must contend with are ethics, data privacy, and the imperative need for effective collaboration with artificial intelligence (*Akgun & Greenhow, 2022*). As we

**Table 2 Linear regression results.** Results shown present how the "educator level of integration" influences GAI utilization.

**Regression statistics**

| | |
|---|---|
| Multiple R | 0.52 |
| R square | 0.279 |
| Adjusted square | 0.26 |
| Standard error | 1.24 |

**ANOVA**

| | df | SS | MS | F | Significance F |
|---|---|---|---|---|---|
| Regression | 1 | 40.522 | 40.52 | 26.03 | 2.9719E−06 |
| Residual | 67 | 104.28 | 1.554 | | |
| Total | 68 | 144.81 | | | |

| | Coefficients | Standard error | t Stat | *p*-value | Lower 95% | Upper 95% |
|---|---|---|---|---|---|---|
| Intercept | 1.575 | 0.420 | 3.749 | 0.00033 | 0.732 | 2.41 |
| GAI utilization | 0.628 | 0.123 | 5.1165 | 2.97E−06 | 0.389 | 0.847 |

advance in integrating AI into education, it is crucial to exercise caution, ensuring that the advantages of AI are fully harnessed while conscientiously addressing any potential challenges (*Kang, Li & Sohaib, 2023*; *Alammari, Sohaib & Younes, 2022*; *Baker, Smith & Anissa, 2019*).

Academics' favorable stance toward GAI contradicts specific research findings on incorporating technology in education. A systematic review conducted by *Celik et al. (2022)* indicated that educators typically need more support to adopt emerging technologies in their teaching due to the complex nature and wide variety of these new tools. The contrasting outcomes observed in our present study may be attributed to the fact that ChatGPT faces fewer external obstacles. Notably, over 50% of the participants we surveyed had gained some experience with ChatGPT in less than 6 months after its launch in November 2022 (*Lampropoulos, Ferdig & Kaplan-Rakowski, 2023*). Furthermore, the rapid rise and widespread adoption of ChatGPT have motivated educators to closely evaluate this AI tool, as noted by *Firat (2023)* and *Lampropoulos, Ferdig & Kaplan-Rakowski (2023)*. The frequent media coverage of ChatGPT and other AI tools, along with their swift progress, may lead to increased utilization and integration by educators.

The findings suggest a positive connection between teachers' awareness of GAI and their usage, aligning with the research of Kaplan-Rakowski and others in 2023. This correlation supports prior research emphasizing the link between teachers' exposure to AI, their trust in the technology (as demonstrated by *Nazaretsky et al. (2022)*), and their willingness to incorporate AI into their teaching practices (as shown by *Kuleto et al. (2022)*). The findings also showed that most participants already contemplate specific AI applications (representing the understanding stage) or feel comfortable using AI (reflecting the familiarity stage), increasing awareness and utilization. It's reasonable to expect a shift

towards greater integration levels over time, specifically in the adaptation and creative application stages.

Teachers' perceptions of AI support the reported understanding and familiarity stages of integration as an instructional tool and their expectations of easy implementation. In these stages, teachers actively consider how to confidently employ GAI for specific tasks. The perceived value and comfort of integrating GAI, like ChatGPT, into educational settings contribute to the positive attitudes observed. ChatGPT is web-based and easily accessible through account creation without additional equipment. It appears that teachers are ready to embrace GAI, which is a welcome departure from earlier studies where teachers often expressed unpreparedness for AI integration, as found in the studies by *An et al. (2022)*, *Alharbi & Sohaib (2021)*, *Celik et al. (2022)* and *Nazaretsky et al. (2022)*.

The study reveals that Generative Artificial Intelligence (GAI) holds transformative educational potential, offering personalized learning experiences and streamlining administrative tasks. Policymakers and educators should acknowledge the positive impact and address ethical, data privacy, and practical collaboration challenges. The link between teachers' awareness and usage of GAI suggests a promising trend for integration, requiring supportive policies.

## Qualitative insights

The interviews with participants shed light on a spectrum of viewpoints and experiences, offering valuable qualitative insights into incorporating GAI in education. A common theme emerged regarding the dual nature of GAI, encapsulated in P1's statement, which found both benefits and drawbacks in its use. While acknowledging its incredible utility, concerns surfaced regarding its potential for facilitating plagiarism, especially at the university level. On the other hand, P2's perspective was notably optimistic, emphasizing the substantial improvement in academic performance due to GAI's introduction. This participant attributed the success to personalized assistance and feedback, which enhanced students' understanding and improved grades. P3 voiced concerns about students leaning heavily on GAI at the expense of traditional resources like library books, emphasizing the importance of striking a balance and encouraging the use of both sources. P3's viewpoint mirrored a cautious stance.

A contrasting perspective emerged from a different P1, highlighting the role of GAI in fostering collaborative learning. Here, students' engagement in projects and knowledge sharing was seen as a positive outcome of GAI implementation, fostering teamwork and collaboration. P2 shared their personal growth due to GAI integration, emphasizing its impact on their professional development. It forced them to adapt, embrace new technology, and explore innovative teaching methods, leading to a fulfilling and intellectually stimulating journey. Regarding implications, P3 expressed concerns about the potential reduction in teaching positions due to increased GAI use. They underscored the irreplaceable role of human educators in providing guidance and mentorship, advocating for a cautious approach to striking the right balance. The participants' responses collectively portray a nuanced and multifaceted landscape of GAI's role in education. While P1 expresses optimism, P2 offers a balanced perspective, and P3

underscores caution. The interviews emphasize the need for a comprehensive understanding of the multifaceted impact of GAI in education, encompassing academic outcomes, concerns of overreliance, and promoting collaboration among students.

Qualitatively, educators exhibit diverse views on GAI, with some emphasizing its benefits and others expressing caution. Policymakers should consider nuanced guidelines to balance traditional and AI-based sources, promoting comprehensive integration. Educators' positive attitudes indicate readiness, suggesting the need for policies supporting responsible GAI adoption in education. In conclusion, a more detailed discussion with specific recommendations would enhance the study's contribution to guiding future GAI integration policies and practices in education.

## CONCLUSION

The convergence of quantitative and qualitative findings offers a comprehensive perspective on integrating GAI in education. Both sets of data contribute valuable insights into the multifaceted impact of this technology on educators and learners. The quantitative data, primarily derived from surveys and statistical analyses, illuminates essential trends and patterns. Increased awareness of GAI correlates with more frequent utilization among teachers, with trust and confidence playing pivotal roles. This aligns with prior research indicating that exposure to AI technologies fosters trust and integration into educational practices. Approximately half of the surveyed educators already find themselves at the understanding and familiarity stages of GAI integration, suggesting a readiness for its adoption. Moreover, the perceived value and ease of GAI assimilation among educators encouraged and likely to incorporate GAI into their teaching methods. GAI tools' user-friendly and web-based nature, like ChatGPT, enhances their accessibility and implementation.

The qualitative analysis of interview responses complements the quantitative findings by offering a deeper understanding of individual experiences and perceptions. These interviews revealed a diversity of viewpoints and experiences among educators. While some, like P1, expressed optimism and enthusiasm for GAI's potential in enhancing academic achievement, others, like P2, approached the technology cautiously, recognizing its benefits but emphasizing potential drawbacks. P3 highlighted the need to balance traditional and AI-based resources and expressed concerns about students needing to catch up on GAI. Notably, the interviews unveiled a range of experiences, from GAI serving as a catalyst for collaborative learning to fostering professional growth, as emphasized by P2.

In conclusion, the convergence of quantitative and qualitative findings underscores the complex and evolving landscape of GAI integration in education. While quantitative data reveal the trends in awareness and adoption, qualitative insights emphasize individual nuances and concerns. GAI offers promising opportunities for improving academic achievement, fostering collaboration, and encouraging professional development among educators. However, challenges such as potential overreliance, plagiarism concerns, and the need for a balanced approach have also come to the fore. The results highlight the importance of careful consideration and ongoing research to strike the right balance in

harnessing GAI's potential in education. Ultimately, the findings suggest that educators are ready to embrace GAI, but it is essential to navigate this integration thoughtfully, recognizing the duality of its impact on teaching and learning.

### Limitations and future work

This study acknowledges several limitations. Firstly, it predominantly centers on the viewpoints and encounters of educators. A more encompassing understanding of GAI integration in education could be achieved by broadening the scope to include student and stakeholder perspectives. Moreover, the research relies on cross-sectional data, and a longitudinal approach could provide insights into the evolution of GAI adoption over time. Furthermore, the study needs to delve deeper into the specific types of GAI tools or applications, which could vary significantly in their impact on education. Acknowledging these limitations is essential for interpreting the findings and guiding future research in this evolving field.

### Funding

The authors received no funding for this work.

### Competing Interests

The authors declare that they have no competing interests.

### Author Contributions

- Abdullah Alammari conceived and designed the experiments, performed the experiments, analyzed the data, performed the computation work, prepared figures and/or tables, authored or reviewed drafts of the article, and approved the final draft.

### Data Availability

The data is available in the Supplemental File.

### Supplemental Information

Supplemental information for this article can be found online at http://dx.doi.org/10.7717/peerj-cs.1879#supplemental-information.

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
