# Peer review of "Evaluating generative AI integration in Saudi Arabian education: a mixed-methods study"

_PeerJ Computer Science, doi:10.7717/peerj-cs.1879_

## Round 0.1 · original submission · Major Revisions

Dear author
Your paper has been reviewed by the experts in the field with interest, I have also evaluated it, and based on the input you will see that the paper needs major improvements as mentioned by the experts, please also specifically address the following concerns along with others

1. Improve the quality of abstract
2. The technical language should be improved
3. Justify and validate the novelty of your proposed research.
Please revise carefully and resubmit.

**Language Note:** The Academic Editor has identified that the English language must be improved. PeerJ can provide language editing services - please contact us at copyediting@peerj.com for pricing (be sure to provide your manuscript number and title). Alternatively, you should make your own arrangements to improve the language quality and provide details in your response letter. – PeerJ Staff

·

Basic reporting

The research focuses on exploring the effects of General Artificial Intelligence (GAI) in education, employing both quantitative and qualitative approaches. The study investigates Saudi educators' perspectives on integrating GAI into the Saudi education system and examines whether educators' familiarity with GAI predicts its frequency of use in educational settings. Overall, the study is conducted well. However, below is the feedback to improve the paper further.

Experimental design

The study utilizes a mixed-method approach, combining surveys and interviews to collect quantitative and qualitative data from Saudi educators. The online survey draws on validated instruments from previous research, though the report needs explicit information on the participant sample size, potentially impacting the generalizability of results. Furthermore, focusing on educators who have used ChatGPT may limit the study's representativeness.

Validity of the findings

The authors should explain a potential issue that arises from adapting a survey instrument without clear validation for the Saudi context. Cultural and contextual differences may affect participant responses, influencing the overall validity of the results.

Additional comments

The study overlooks ethical considerations related to GAI integration in education, such as privacy, data security, and bias. It also needs to assess the impact of GAI on student learning outcomes.

Reviewer 2 ·

Basic reporting

This study employs a mixed-method approach, combining quantitative data collection through surveys and qualitative data collection through interviews to gather insights from educators in Saudi institutions. The following points need to be included in the study.

• In methodology, the study could provide information on how participants were selected and recruited and the criteria used to determine eligibility.
• Participants' profile information from the Interview should also be presented.
• The study could describe the statistical methods used to analyze the quantitative data and the coding process used to analyze the qualitative data.
• The study could provide more examples of the themes and subthemes that emerged from the interviews and the specific quotes from participants that support these themes. Additionally, the study could use a more rigorous coding process, such as inter-coder reliability testing, to ensure the validity and reliability of the qualitative data analysis.
• A more thorough discussion of the practical implications for policymakers and educators, including specific recommendations, would enhance the study's contribution to guiding future policies and practices in GAI integration in education.

Experimental design

As above

Validity of the findings

As above

Additional comments

As above

Reviewer 3 ·

Basic reporting

1- The author should include some visuals in the introduction section to make it appealing.
2- The author should increase the introduction section, including some more details about the problem.
3- The author should include some recent paper in the literature review section.
4- The author should add a comparison table or summary paragraph at the end of the literature review highlighting the key findings in previous literature.

Experimental design

5- The author should include an overall methodology figure in the relevant section.

Validity of the findings

6- The author should try to compare the results with some of the relevant studies.
7- The author should include some future work in the conclusion section.
8- There is no visual in the paper, the author should look to add some figures to increase the readability of paper.

---

## Round 0.2 · accepted · Accept

The reviewers are happy with the revised version of your paper, therefore we are pleased to inform you that your manuscript has been recommended for publication. Thanks for your contribution.

·

Basic reporting

All my previous comments have been addressed in this final version.

Experimental design

All my previous comments have been addressed in this final version.

Validity of the findings

All my previous comments have been addressed in this final version.

Additional comments

None

Reviewer 2 ·

Basic reporting

In the revised version the authors carefully address the comments. So the current version is acceptable for publication.

Experimental design

In the revised version the authors carefully address the comments. So the current version is acceptable for publication.

Validity of the findings

In the revised version the authors carefully address the comments. So the current version is acceptable for publication.

Reviewer 3 ·

Basic reporting

All changes have been completed.

Experimental design

All changes have been completed.

Validity of the findings

All changes have been completed.